

# Perceived fear and exercise difficulty in patients with migraine and their association with psychosocial factors: a cross-sectional study

Álvaro Reina-Varona[1,2,3], Beatriz Madroñero-Miguel[2], Alba Paris-Alemany[1,4,5] and Roy La Touche[1,2,4]

[1] Motion in Brains Research Group, Centro Superior de Estudios Universitarios La Salle, Universidad Autónoma de Madrid, Madrid, Spain
[2] Departamento de Fisioterapia, Centro Superior de Estudios Universitarios La Salle, Universidad Autónoma de Madrid, Madrid, Spain
[3] Facultad de medicina, Universidad Autónoma de Madrid, Madrid, Spain
[4] Instituto de Dolor Craneofacial y Neuromusculoesquelético (INDCRAN), Madrid, Spain
[5] Department of Basic Health Sciences, Universidad Rey Juan Carlos, Alcorcón, Madrid, Spain

## ABSTRACT

**Purpose:** Although pharmacological treatments for migraine have advanced, non-pharmacological approaches, such as exercise, offer additional benefits. However, many patients avoid physical activity due to fear of symptom exacerbation. This study aims to identify the most threatening and difficult exercises for patients with migraine and examine the relationship between exercise perceptions and factors such as physical activity levels, kinesiophobia, catastrophizing, and self-efficacy.

**Methods:** This cross-sectional study explored the perceived fear and difficulty of nine exercises shown *via* video demonstrations, and their association with psychological variables in patients with migraine. Participants aged 18–65 with physician-diagnosed migraines completed self-report measures of physical activity, kinesiophobia, catastrophizing, fear-avoidance beliefs, headache impact, neck disability, and self-efficacy.

**Results:** A total of 110 patients with migraine participated (88% women), with a mean age of 36 years. Chronic migraine was reported by 53% of participants. Significant differences in perceived fear and difficulty were observed across the nine exercises ($p < 0.001$), with jumping and spine extension rated as the most fear-inducing and difficult exercises, respectively. Beta regression models showed that the physical activity level, measured by the short form of the International Physical Activity Questionnaire (IPAQ-SF), was negatively associated with fear of jumping and difficulty of spine extension, whereas fear-avoidance beliefs were positively associated with both. A correlation analysis revealed a moderately significant negative association between the IPAQ-SF score and perceived fear of running.

**Conclusion:** Jumping, running, and spine extension were perceived as the most aversive exercises. Perceived fear and difficulty levels were associated with physical activity levels and fear avoidance beliefs, emphasizing the need to assess these factors before implementing therapeutic exercise interventions.

Corresponding author
Alba Paris-Alemany,
albaparis@gmail.com

# INTRODUCTION

Migraine implies a complex neurological disorder in which a peripheral and central sensitized trigeminovascular system interacts, producing the most disabling headache condition in the world (*Goadsby et al., 2017*; *Vos et al., 2017*). The high prevalence of migraine, its impact on patients' quality of life, and the elevated costs associated with its treatment and work absenteeism have positioned it as a relevant target for the development of pharmacological interventions (*Evers et al., 2009*; *Linde et al., 2012*; *Vos et al., 2017*; *Stovner et al., 2018*; *Buse et al., 2019*). These approaches have yielded good results, especially with new-generation treatments, such as calcitonin gene-related peptide pathway monoclonal antibodies and gepants (*Lattanzi et al., 2019*; *Yang et al., 2021*). However, non-pharmacological strategies should be considered as adjuvant interventions due to their beneficial effects on improving migraine symptoms and quality of life (*La Touche et al., 2020*; *Wu et al., 2022*).

In cohort studies, higher physical activity levels in patients with this condition have been associated with fewer monthly migraine episodes (*Seok, Cho & Chung, 2006*; *Woldeamanuel & Cowan, 2016*; *Hagan et al., 2021*). Indeed, patients with transformed migraine who more effectively engaged in preventive medication use and regular exercise reduced their continuous use of analgesics and transitioned from transformed to episodic migraine (*Seok, Cho & Chung, 2006*). Various exercise modalities have been shown to reduce migraine frequency, duration, intensity, and disability (*Lemmens et al., 2019*; *La Touche et al., 2020*, *2023*; *Wu et al., 2022*). Moderate-continuous aerobic exercise and yoga are the most studied exercise interventions, although other modalities, such as resistance training, have begun to be explored (*La Touche et al., 2023*; *Reina-Varona et al., 2024*). Moreover, emerging evidence has shown that dual task based on the combination of exercise and cognitive tasks could effectively improve cognition and disability in patients with migraine (*Deodato et al., 2024*). Combining exercise interventions with lifestyle recommendations, such as regular sleep, mealtimes, and hydration, could increase the health status of these patients (*La Touche et al., 2023*).

Despite the current evidence, patients with migraine appear to avoid physical activity, even during headache-free days, compared with healthy controls (*Rogers et al., 2020*). Avoidance of physical activity has been correlated with lower levels of vigorous physical activity and a stronger expectation that physical activity could trigger and worsen migraines and promote more frequent migraine episodes (*Farris et al., 2019*). Furthermore, migraine worsening related to physical activity was associated with more severe nausea, photophobia, phonophobia, and allodynia (*Farris et al., 2018*).

Evidence has indicated that patients with more severe migraine symptoms tend to engage in less physical activity, although the relationship between these factors remains unclear. Given the positive results observed in studies, patients with migraines could benefit from implementing exercise programs (*La Touche et al., 2020*, *2023*; *Wu et al., 2022*). However, the fear of triggering or worsening a migraine episode might influence the

low adherence observed among patients with more severe symptoms. Consequently, the impact of psychological factors, such as kinesiophobia, should be considered. Kinesiophobia and avoidance behaviors have been associated with greater disability, pain catastrophizing, hypervigilance, depression, and anxiety scores, and lower levels of self-efficacy and physical activity time in patients with migraine (*Pina et al., 2024*; *Toprak Celenay et al., 2024*). Other relevant factors that could affect the success of exercise in migraine treatment include pain catastrophism and self-efficacy. Migraine patients often present clinically relevant levels of pain catastrophizing, notably higher in chronic than episodic migraine (*Pistoia et al., 2022*). Pain catastrophizing has also been associated with greater levels of disability in this population (*Alvarez-Astorga et al., 2021*). Conversely, higher levels of self-efficacy and exercise engagement have been linked to lower migraine-related disability (*Woldeamanuel et al., 2020*).

To enhance exercise adherence among patients with migraines, it is necessary to deepen our understanding of the barriers and beliefs they hold toward exercise, as well as the potential psychosocial obstacles they face. Therefore, the objectives of the present study were to identify which of nine exercises (walking, running, jumping, neck rotation, neck flexion/extension, spine flexion and extension, squatting, and shoulder press) were perceived as the most threatening and difficult, and to evaluate the association between physical activity levels, kinesiophobia, catastrophizing, fear-avoidance beliefs, self-efficacy, and headache and neck disability with the perceived fear and difficulty of said exercises.

## MATERIALS AND METHODS

### Study design

An observational cross-sectional study was conducted from March 2022 to September 2024 to evaluate the perceived fear and difficulty reported by patients with migraine while observing various exercises and the association of these scores with psychological variables. We used non-probabilistic convenience and snowball sampling, employing social networks to spread the study information. The study was performed according to the Strengthening the Reporting of Observational Studies in Epidemiology declaration (*von Elm et al., 2008*). It followed the principles of the Declaration of Helsinki and obtained the approval of the Ethics Committee of the University La Salle (CSEULS-PI-005/2022) on the 3rd of February 2022. All participants gave written informed consent to participate.

### Participants

Patients with episodic or chronic migraines, with or without visual or motor aura, were included. They could have comorbidities commonly associated with migraine, such as tensional headache, irritable bowel syndrome, or fibromyalgia.

Patients were included if they (1) were 18 to 65 years of age; (2) had physician-diagnosed migraine; (3) had their migraine diagnosis corroborated by the Penn Online Evaluation of Migraine (POEM) questionnaire; and (4) were fluent in Spanish. Patients were excluded if they: (1) had a history of cervical, facial, or cranioencephalic

trauma that could have started the headache; or (2) were pregnant. Severe comorbidities, such as undergoing cancer treatment, were also considered for exclusion.

## Procedure

First, patients received an e-mail containing links to a video call and the questionnaires, which were adapted online with Cognito Forms (https://www.cognitoforms.com/), a free online form builder. Upon opening the first link and reading the informed consent, they gave their consent by clicking the "yes" button. Next, patients confirmed the inclusion criteria and migraine diagnosis by completing the POEM questionnaire. The evaluator always supervised this process during the video call to guide patients through the test and to address any questions that arose. After confirming the inclusion criteria and migraine diagnosis, participants proceeded to the next part of the evaluation.

The second part consisted of various sociodemographic and anthropometric questions (age, height, weight, academic level, and work status) and included inquiries about other health conditions, such as additional headache diagnoses (tensional headache, cluster headache, hemicrania continua, or new persistent daily headache) or comorbidities (cardiovascular diseases, type 2 diabetes mellitus, rheumatoid arthritis, or fibromyalgia). Following these questions, participants completed the Spanish versions of the short form of the International Physical Activity Questionnaire (IPAQ-SF), the shortened version of the TAMPA Scale of Kinesiophobia (TSK-11), the Pain Catastrophizing Scale (PCS), the Fear-Avoidance Beliefs Questionnaire (FAB-Q), the Headache Impact Test (HIT-6), the Neck Disability Index (NDI), the Chronic Pain Self-Efficacy Scale (CPSES), and the Self-Efficacy Questionnaire to Regulate Exercise (SEQRE).

Lastly, the third part involved assessments of nine different exercises (walking, running, jumping, neck rotation and flexion/extension, spine flexion and extension, squatting, and shoulder press) to evaluate the perceived fear of initiating or aggravating a migraine episode and the difficulty of performing the exercise on a scale from 1 to 10. Given the differences in perception between interictal and migraine phases, patients were advised to consider both moments and weigh their final scores accordingly. The selection of these exercises was based on their common inclusion as aerobic (walking, running, and jumping), mobility (neck rotation, flexion, and extension, and spine flexion), and strength exercises (spine extension, squatting, and shoulder press) in conventional exercise and rehabilitation programs. The evaluator showed videos of these exercises to the participants to facilitate the scoring process (available in Material S1). In each video, a female demonstrator briefly performed each exercise to ensure clarity on what was being described. After viewing the videos, participants rated their perceived fear and difficulty for each exercise.

## Outcomes

The POEM questionnaire is a validated instrument for confirming migraine diagnosis, with specificities of 84%, 93%, and 90% for migraine without aura, migraine with aura, and overall migraine diagnosis, respectively (Kaiser et al., 2019).

The Spanish version of the IPAQ-SF was used to evaluate each participant's physical activity level. This questionnaire is valid and reliable, showing moderate concordance between both long and short forms (r = 0.67; 95% CI [0.64–0.7]) and reliability (r = 0.76; 95% CI [0.73–0.77]) (*Toloza & Gómez-Conesa, 2007*). It includes seven items that evaluate the days per week and minutes per day of vigorous, moderate, and low physical activity (walking), as well as the minutes per day spent sitting. The physical activity level is then calculated in metabolic equivalents for vigorous, moderate, and low intensities, and for total physical activity.

Kinesiophobia, a construct that assesses fear of pain and movement, was evaluated with the Spanish version of the TSK-11, which is composed of two subscales: fear of physical activity and fear of injury. This is a valid and reliable instrument (Cronbach's α = 0.79; intraclass correlation coefficient (ICC) > 0.7) (*Gómez-Pérez, López-Martínez & Ruiz-Párraga, 2011*).

Catastrophism was assessed with the Spanish version of the PCS. This scale contains three domains: rumination, magnification, and hopelessness. It is a valid and reliable instrument (Cronbach's α = 0.79; ICC = 0.84) (*Campayo et al., 2008*). The authors have permission to use this instrument from the copyright holders.

The FAB-Q was employed to evaluate fear-avoidance beliefs and behaviors related to physical activity and work. It is a valid and reliable instrument (Cronbach's α = 0.93; ICC = 0.97) (*Kovacs et al., 2006*). However, items 3 and 11 were eliminated because they evaluate the perception that physical activity/work might harm the back. A reliability analysis was conducted after recruitment to ensure high internal consistency, even if these items were removed.

Headache-related disability was assessed using the HIT-6 questionnaire, which has a high internal consistency (Cronbach's α = 0.87) (*Martin et al., 2004*). It consists of six questions related to pain severity, limited daily activities, frequency of wishing to lie down, tiredness to do work and daily activities, frequency of feeling irritated, and limited concentration.

Neck pain is a common comorbidity among patients with migraine (*Ashina et al., 2015*). Therefore, the Spanish version of the NDI questionnaire was included to evaluate the disability generated by neck pain. This version is a valid and reliable instrument (Cronbach's α = 0.89; ICC = 0.98) (*Kovacs et al., 2008*; *Ortega, Martínez & Ruiz, 2010*). The authors have permission to use this instrument from the copyright holders.

The self-efficacy construct was assessed with the Spanish version of the CPSES. This questionnaire contains three domains: self-efficacy for pain management, physical functioning, and coping with symptoms. The validity of this instrument is good (Cronbach's α = 0.91), and it is reliable (ICC = 0.75) (*Martín-Aragón et al., 1999*). The authors have permission to use this instrument from the copyright holders.

Lastly, the short form of the SEQRE was used to evaluate the perceived self-efficacy to maintain a daily exercise routine. The Spanish version shows good internal consistency (Cronbach's α = 0.84) (*de los Fuentes Vega & González Lomelí, 2020*). It contains seven different contexts that could be interpreted as barriers to exercise adherence, and each item is scored from 0 ('I cannot do it') to 100 points ('I am confident I can do it').

## Sample size calculation

A sample size calculation was made based on a multiple linear regression evaluating the association between the analyzed variables and the perceived fear of jumping. The sample size required in this study for the results of this regression analysis was calculated using an alpha level of 0.05 and a 90% power. We calculated the effect size ($f^2 = 0.19$) based on the adjusted coefficient of determination ($R^2 = 0.16$). These parameters and results were introduced into the G*Power 3.1 software (University of Düsseldorf, Düsseldorf, Germany) (*Faul et al., 2007*). A total sample size of 109 participants was estimated based on these parameters.

## Statistical analysis

All statistical analyses were performed with R software (version 4.4.1; *R Core Team, 2024*) in the RStudio environment (*RStudio Team, 2023*) version 2023.06.0+421. Descriptive analyses were conducted to summarize the demographic and clinical characteristics of participants, including age, height, weight, body mass index (BMI), sex, academic level, employment status, and migraine characteristics. Mean, standard deviation, medians, minimum, maximum, and quartiles 1 and 3 were reported for continuous variables. Categorical variables were summarized using frequencies and percentages.

The primary aim was to assess differences in perceived fear and difficulty across nine exercises. Given the nature of the data, which consisted of bounded ordinal variables, robust statistical methods were applied, using the WRS and WRS2 packages (*Mair & Wilcox, 2019*; *Wilcox, 2021*). We used a robust one-way analysis of variance (ANOVA) for medians (bd1way function), with 2,000 bootstrap resamples, given the non-normality of the data. Multiple comparisons of medians between exercises were conducted using the percentile bootstrap method (dmedpb function). Results are presented as median difference values along with their 95% percentile bootstrap confidence interval and interquartile ranges.

For the secondary aim, beta regression models were employed to examine the relationships between the total scores of the various questionnaires (IPAQ-SF, TSK-11, PCS, FAB-Q, HIT-6, NDI, CPSES, and SEQRE) and the perceived fear and difficulty during each exercise. Only those exercises with the highest scores for fear and difficulty were selected for the beta regression analyses. These scores were standardized to values between 0 and 1 and were analyzed with the betareg package. Specifically, the distribution of these standardized values was examined to ensure they were bounded between 0 and 1 (*Ferrari & Cribari-Neto, 2004*). Transformations were applied to ensure that values which reached exactly 0 or 1 were shifted slightly (*e.g.*, 0.0001 and 0.9999) to fit within the bounds required for beta regression.

Before running the beta regressions, the assumptions of the model were evaluated. Model diagnostics such as residual *vs* fitted, Cook's distance, leverage, and Q-Q plot measures were evaluated to check for violations of assumptions.

To interpret significant findings from the beta regression models, predicted values and their 95% prediction intervals were calculated and plotted for each questionnaire score. The relationships between variables were visualized using smooth curves.

We also included two Spearman correlation analyses to evaluate the strength of the association between the perceived fear in different exercises and the questionnaires' scores, and between the perceived difficulty in various exercises and the questionnaires' scores (*Schober, Boer & Schwarte, 2018*).

The internal consistency of the FAB-Q questionnaire was analyzed with a reliability analysis. Both Cronbach's α (0.87) and McDonald's ω (0.88) indicated high internal consistency (results are available in Material S2).

## RESULTS

### Descriptive statistics

A total of 110 patients with migraine participated in the study, with 88% identifying as female. The average age was 36 years (SD ± 12), and 88% had a college-level education. Most (63%) participants were employed, and 25% were students. The cohort's mean BMI was 25 (SD ± 4). Regarding migraine diagnosis, 53% had chronic migraine, 47% had episodic migraine, 30% had migraine without aura, 64% had visual aura symptoms, and 36% had motor aura symptoms. Table 1 shows the descriptive analysis of the participants. Histograms and pairwise correlations are provided in Material S3.

### Primary analysis: perceived fear and difficulty across exercises

The one-way robust ANOVA for median values revealed significant differences in perceived fear across the nine exercises (test = 13.56; $p < 0.001$). Jumping was perceived as the most fear-inducing exercise, with a median value of 5. The differences in perceived fear between the exercises are presented in Fig. 1A.

Similarly, there were significant differences in perceived difficulty among the exercises (test = 8.89; $p < 0.001$). Spine extension was rated as the most difficult, with a median score of 4. The differences in perceived difficulty between the exercises are presented in Fig. 1B.

### Secondary analysis: regression models for perceived fear and difficulty

Beta regression models were employed to assess the relationship between perceived fear of jumping and the various questionnaire scores, and between perceived difficulty of spine extension and the questionnaire scores. The assumptions of the beta regression models were examined. Although Cook's distance showed no significant influence of each score in the various models, some participants appeared to influence the models' estimation based on the leverage plot. However, there was no justification for their elimination. Some deviations from the half-normal plot of residuals were also presented. The assumptions of the beta regression models are included in Material S4.

**Table 1 Descriptive data of the study sample.**

| Variable | N (Total = 110) | % | Mean | Std. Dev. | Min | Q1 | Median | Q3 | Max |
|---|---|---|---|---|---|---|---|---|---|
| Migraine frequency | | | | | | | | | |
| *Chronic* | 58 | 53% | | | | | | | |
| *Episodic* | 52 | 47% | | | | | | | |
| Aura | | | | | | | | | |
| *No* | 33 | 30% | | | | | | | |
| *Visual* | 70 | 64% | | | | | | | |
| *Motor* | 40 | 36% | | | | | | | |
| Sex | | | | | | | | | |
| *F* | 97 | 88% | | | | | | | |
| *M* | 13 | 12% | | | | | | | |
| Academic level | | | | | | | | | |
| *Primary* | 3 | 3% | | | | | | | |
| *Secondary* | 10 | 9% | | | | | | | |
| *College* | 97 | 88% | | | | | | | |
| Employment status | | | | | | | | | |
| *Student* | 28 | 25% | | | | | | | |
| *Unemployed* | 10 | 9% | | | | | | | |
| *Active* | 69 | 63% | | | | | | | |
| *Retired* | 3 | 3% | | | | | | | |
| Comorbidities | | | | | | | | | |
| *Tensional-type headache* | 20 | 20% | | | | | | | |
| *Trigeminal autonomic cephalalgias* | 1 | 1% | | | | | | | |
| *Depression* | 5 | 5% | | | | | | | |
| *Anxiety* | 5 | 5% | | | | | | | |
| *Major psychiatric disorders* | 3 | 3% | | | | | | | |
| *POS* | 5 | 5% | | | | | | | |
| *Musculoskeletal disorders* | 6 | 5% | | | | | | | |
| *Hypo/hyperthyroidism* | 7 | 6% | | | | | | | |
| *Asthma* | 5 | 5% | | | | | | | |
| *Psoriasis* | 3 | 3% | | | | | | | |
| *Fibromyalgia* | 2 | 2% | | | | | | | |
| *Gastro-intestinal inflammatory disease* | 8 | 7% | | | | | | | |
| *Cardiovascular disease* | 13 | 12% | | | | | | | |
| *Hematological disease* | 2 | 2% | | | | | | | |
| *Diabetes Type II* | 3 | 3% | | | | | | | |
| *Neurological disease* | 5 | 5% | | | | | | | |
| *Visual disorders* | 3 | 3% | | | | | | | |
| Age (y) | | | 36 | 12 | 18 | 26 | 32 | 45 | 64 |
| Height (cm) | | | 166 | 7 | 150 | 162 | 166 | 170 | 188 |
| Weight (kg) | | | 68 | 13 | 46 | 59 | 66 | 75 | 112 |
| BMI (Kg/m$^2$) | | | 25 | 4 | 17 | 21 | 24 | 27 | 39 |

| Variable | N (Total = 110) | % | Mean | Std. Dev. | Min | Q1 | Median | Q3 | Max |
|---|---|---|---|---|---|---|---|---|---|
| IPAQ-SF vigorous | | | 872 | 1,070 | 0 | 0 | 480 | 1,440 | 6,000 |
| IPAQ-SF moderate | | | 552 | 911 | 0 | 0 | 240 | 720 | 5,760 |
| IPAQ-SF walking | | | 1,149 | 954 | 0 | 516 | 792 | 1,386 | 4,158 |
| IPAQ-SF total | | | 2,573 | 2,003 | 0 | 1,045 | 2,056 | 3,746 | 8,532 |
| IPAQ-SF category | | | | | | | | | |
| *Sedentary* | 16 | 15% | | | | | | | |
| *Moderate* | 52 | 46% | | | | | | | |
| *High* | 43 | 39% | | | | | | | |
| TSK-11 total | | | 23 | 7.1 | 11 | 16 | 23 | 27 | 42 |
| PCS total | | | 21 | 13 | 0 | 10 | 20 | 32 | 49 |
| FAB-Q total | | | 27 | 17 | 0 | 14 | 23 | 39 | 80 |
| HIT-6 total | | | 64 | 6.5 | 48 | 60 | 64 | 69 | 78 |
| NDI total | | | 11 | 7.9 | 1 | 4 | 9 | 15 | 38 |
| CPSES total | | | 124 | 34 | 17 | 104 | 128 | 146 | 188 |
| SEQRE total | | | 346 | 176 | 0 | 210 | 350 | 480 | 700 |
| Walking fear | | | 1.9 | 1.7 | 1 | 1 | 1 | 2 | 9 |
| Walking difficulty | | | 1.9 | 1.6 | 1 | 1 | 1 | 2 | 8 |
| Running fear | | | 4.2 | 3.1 | 1 | 1 | 4 | 7 | 10 |
| Running difficulty | | | 4 | 2.7 | 1 | 2 | 3 | 6 | 10 |
| Jumping fear | | | 4.5 | 3 | 1 | 1 | 5 | 7 | 10 |
| Jumping difficulty | | | 3.7 | 2.8 | 1 | 1 | 3 | 6 | 10 |
| Neck rotation fear | | | 2.9 | 2.5 | 1 | 1 | 2 | 4 | 10 |
| Neck rotation difficulty | | | 2.4 | 2.2 | 1 | 1 | 1 | 3 | 10 |
| Neck flexion/extension fear | | | 3.2 | 2.7 | 1 | 1 | 2 | 4 | 10 |
| Neck flexion/extension difficulty | | | 2.6 | 2.4 | 1 | 1 | 1.5 | 3 | 10 |
| Spine flexion fear | | | 3.7 | 2.8 | 1 | 1 | 3 | 6 | 10 |
| Spine flexion difficulty | | | 3.1 | 2.5 | 1 | 1 | 2 | 5 | 10 |
| Spine extension fear | | | 4.5 | 3 | 1 | 1.2 | 4 | 7 | 10 |
| Spine extension difficulty | | | 4.4 | 2.8 | 1 | 2 | 4 | 7 | 10 |
| Squatting fear | | | 3 | 2.5 | 1 | 1 | 2 | 4 | 10 |
| Squatting difficulty | | | 3.5 | 2.4 | 1 | 1 | 3 | 5 | 10 |
| Shoulder press fear | | | 2.7 | 2.3 | 1 | 1 | 2 | 4 | 10 |
| Shoulder press difficulty | | | 3.2 | 2.2 | 1 | 1 | 3 | 4 | 10 |

**Note:**
BMI, Body mass index; CPSES, chronic pain self-efficacy scale; F, female; FAB-Q, fear-avoidance beliefs questionnaire; HIT-6, headache impact test; IPAQ-SF, international physical activity questionnaire-short form; M, male; Max, maximum value; Min, minimum value; N, number of participants; NDI, neck disability index; PCS, pain catastrophizing scale; POS, polycystic ovary syndrome; Q1, first quartile; Q3, third quartile; SEQRE, self-efficacy questionnaire to regulate exercise; Std. Dev., standard deviation; TSK-11, TAMPA scale of kinesiophobia.

### Jumping fear

The IPAQ-SF was negatively associated with jumping fear (β = −0.00016, 95% CI [−0.00029 to −0.00003]; $p = 0.013$), whereas the FAB-Q was positively associated (β = 0.03; 95% CI [0.0094–0.0499]; $p = 0.004$). Both associations are shown in Fig. 2.

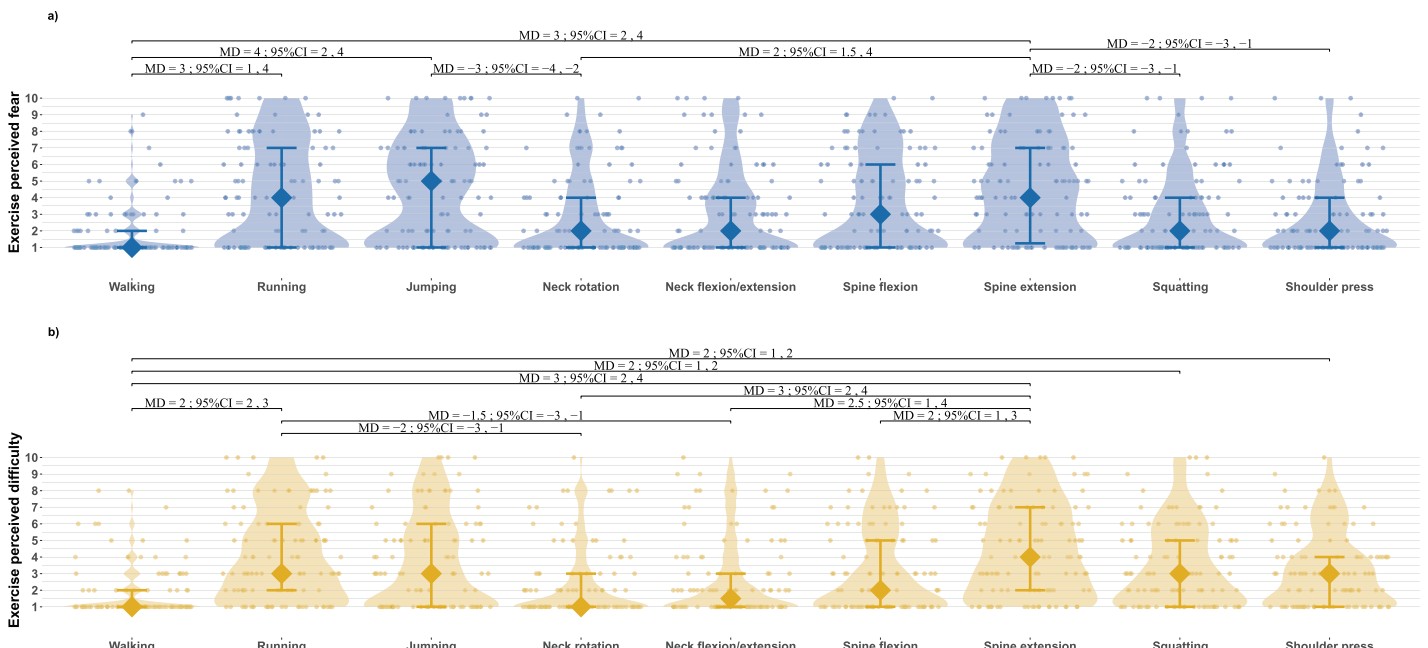

**Figure 1 Median differences between exercises for perceived fear (A) and perceived difficulty (B).** Violin graphics show density curves for frequency of distribution, median values as diamonds, interquartile ranges as bars, and individual data as points. MD, median differences; 95% CI, 95% percentile bootstrap confidence intervals.

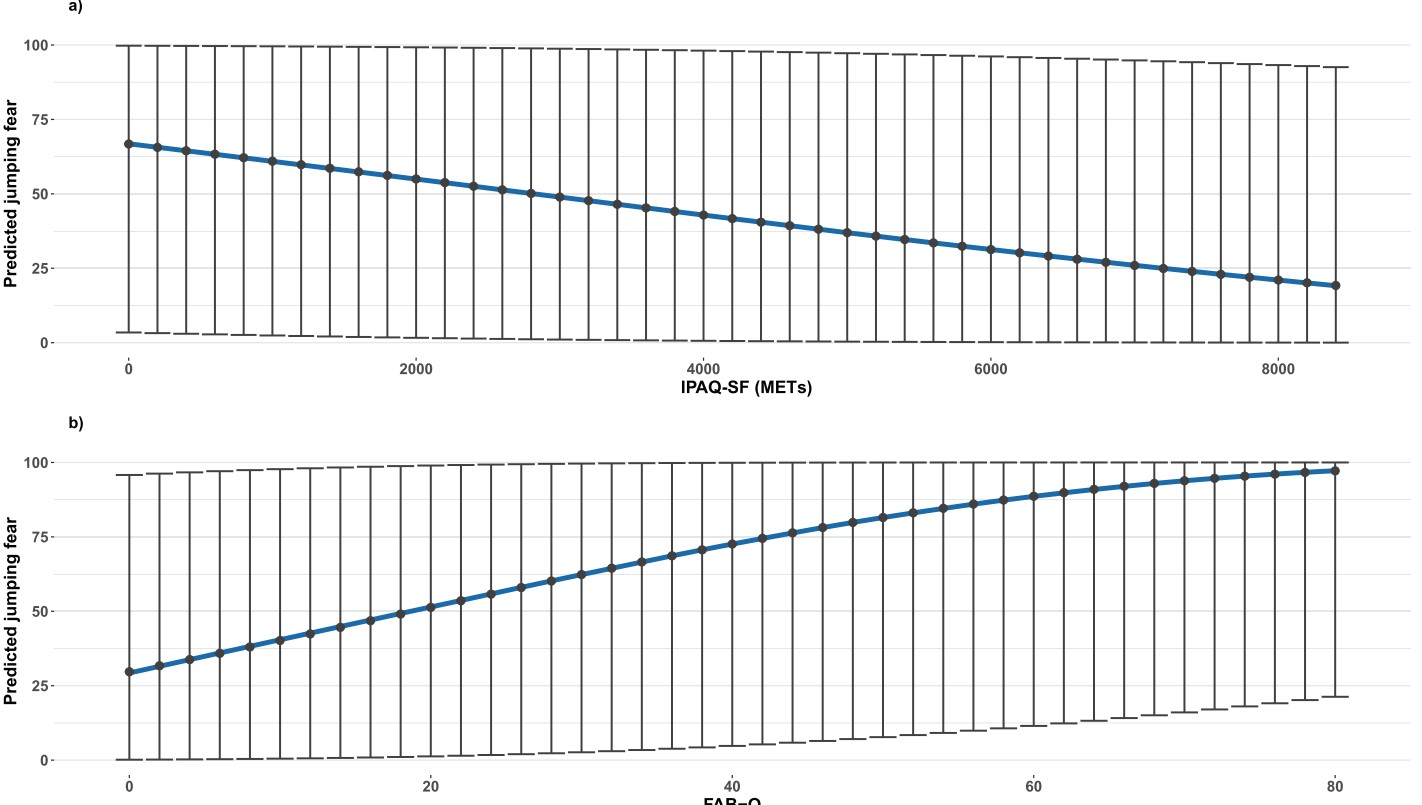

**Figure 2 Beta regression models illustrating the association between jumping fear and IPAQ-SF (A), and FAB−Q (B) scores.** Graphics show predicted median values as points and prediction intervals as bars.

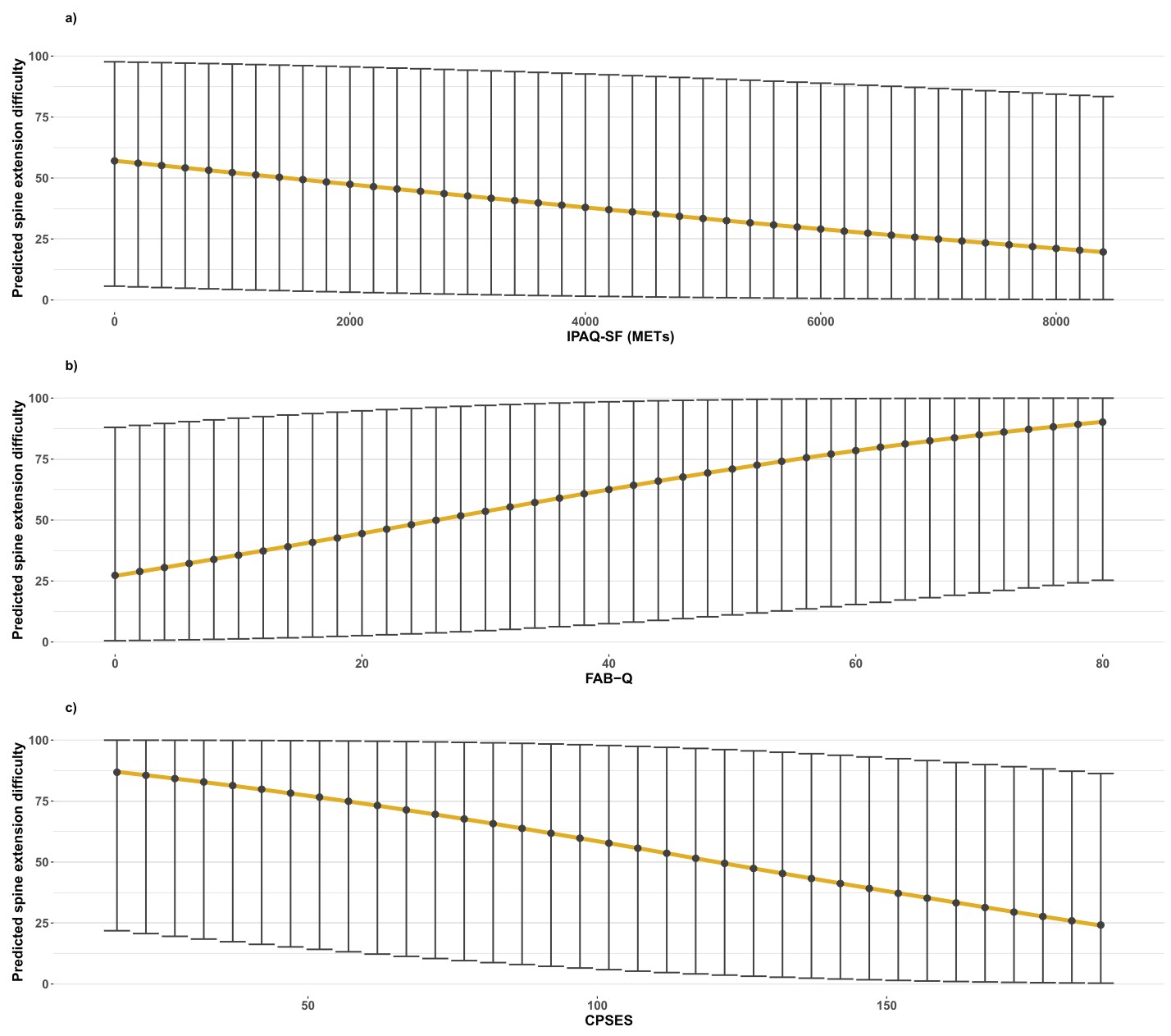

**Figure 3  Beta regression models illustrating the association between lumbar extension difficulty and IPAQ-SF (A), FAB−Q (B), and CPSES (C) scores.** Graphics show predicted median values as points and prediction intervals as bars.

### Spine extension difficulty

Both the IPAQ-SF ($\beta$ = −0.00015; 95% CI [−0.00026 to −0.00003]; $p$ = 0.014) and the CPSES ($\beta$ = −0.0126; 95% CI [−0.0208 to −0.0044]; $p$ = 0.003) scores were negatively associated with spine extension difficulty, whereas the FAB-Q was positively associated ($\beta$ = 0.028; 95% CI [0.0091−0.0462]; $p$ = 0.0035). These associations are presented in Fig. 3.

**Table 2 Correlation analysis of the association between each exercise's perceived fear and the different questionnaires.**

| | \multicolumn{9}{c}{EXERCISE FEAR} | | | | | | | | |
| --- | --- | --- | --- | --- | --- | --- | --- | --- | --- |
| | Walking | Running | Jumping | Neck rotation | Neck flexion/extension | Spine flexion | Spine extension | Squatting | Shoulder press |
| IPAQ-SF | −0.154 | −0.425*** | −0.396*** | −0.249** | −0.252** | −0.337*** | −0.336*** | −0.314** | −0.365*** |
| TSK-11 | 0.121 | 0.213* | 0.178 | 0.209* | 0.203* | 0.175 | 0.211* | 0.236* | 0.32** |
| FAB-Q | 0.189* | 0.373*** | 0.367*** | 0.097 | 0.098 | 0.32** | 0.31** | 0.221* | 0.338*** |
| PCS | 0.145 | 0.172 | 0.157 | 0.123 | 0.087 | 0.207* | 0.12 | 0.22* | 0.262*** |
| NDI | 0.012 | 0.06 | 0.122 | 0.123 | 0.161 | 0.094 | 0.16 | −0.063 | 0.134 |
| HIT-6 | 0.15 | 0.242* | 0.205* | −0.02 | −0.005 | 0.231* | 0.152 | 0.044 | 0.059 |
| CPSES | −0.247** | −0.243* | −0.224* | −0.11 | −0.04 | −0.239* | −0.198* | −0.245* | −0.214* |
| SEQRE | −0.051 | −0.196* | −0.227* | −0.252** | −0.219* | −0.252** | −0.237* | −0.278** | −0.357*** |

Notes:
* $p < 0.05$.
** $p < 0.01$.
*** $p < 0.001$.

**Table 3 Correlation analysis of the association between each exercise's perceived difficulty and the different questionnaires.**

| | \multicolumn{9}{c}{EXERCISE DIFFICULTY} | | | | | | | | |
| --- | --- | --- | --- | --- | --- | --- | --- | --- | --- |
| | Walking | Running | Jumping | Neck rotation | Neck flexion /extension | Spine flexion | Spine extension | Squatting | Shoulder press |
| IPAQ-SF | −0.1 | −0.174 | −0.257** | −0.158 | −0.204* | −0.337*** | −0.382*** | −0.223* | −0.194* |
| TSK-11 | 0.098 | 0.157 | 0.193* | 0.198* | 0.223* | 0.228* | 0.173 | 0.21* | 0.116 |
| FAB-Q | 0.238* | 0.295** | 0.384*** | 0.169 | 0.157 | 0.381*** | 0.387*** | 0.314** | 0.284** |
| PCS | 0.113 | 0.102 | 0.089 | 0.142 | 0.123 | 0.193* | 0.155 | 0.229* | 0.242* |
| NDI | 0.027 | 0.109 | 0.153 | 0.16 | 0.14 | 0.145 | 0.245* | 0.105 | 0.152 |
| HIT-6 | 0.15 | 0.16 | 0.286** | 0.138 | 0.083 | 0.346*** | 0.263** | 0.177 | 0.135 |
| CPSES | −0.228* | −0.254** | −0.257** | −0.146 | −0.118 | −0.321** | −0.346*** | −0.347*** | −0.324** |
| SEQRE | 0.048 | 0.046 | −0.071 | −0.079 | −0.119 | −0.204* | −0.282** | −0.29** | −0.231* |

Notes:
* $p < 0.05$.
** $p < 0.01$.
*** $p < 0.001$.

## Correlation analyses

The results of the fear correlation analysis are summarized in Table 2. The strongest association was between the IPAQ-SF and perceived fear of running, which showed a moderately significant correlation ($\rho = -0.425$, $p < 0.001$). The results of the difficulty correlation analysis are shown in Table 3. No moderate or strong correlations were found in this analysis.

## DISCUSSION

The main objective of the present study was to identify the exercises perceived as most threatening and difficult by a group of patients with migraine while watching a series of videos that included aerobic exercises (walking, running, and jumping), mobility exercises (neck rotation, flexion, and extension, and spine flexion), and strength exercises (spine

extension, squatting, and shoulder press). The secondary objective was to analyze the potential relationships between perceived fear and difficulty in these exercises and scores from various questionnaires assessing physical activity levels, fear of movement, catastrophizing, fear-avoidance beliefs, headache and neck disability, pain, and exercise self-efficacy.

The comparison between exercises revealed that running, jumping, and spine extension obtained the highest scores for both fear and difficulty. Jumping had the highest perceived fear score, whereas spine extension had the highest perceived difficulty. Conversely, walking had the lowest scores for both perceived fear and difficulty. This pattern could primarily be attributed to the varying intensity levels of the exercises. However, squatting and shoulder press, despite receiving lower scores for perceived fear, could also have been perceived as similarly demanding as running, jumping, and spine extension in terms of physical effort.

Some similarities between running, jumping, and spine extension are the impact, jolting, and rapid oscillatory head movements that all entail. Although neck rotation, flexion, and extension exercises also involve head displacement, these movements are smoother and more controlled than those previously mentioned. *Martins, Gouveia & Parreira (2006)* found that 96.3% and 98.1% of patients with migraines experienced an aggravation of migraine pain during jolting and forward bending, respectively. These proportions were significantly higher than those observed in patients with tension-type headaches, of whom only 18.4% and 14.3% reported pain exacerbation with these stimuli (*Martins, Gouveia & Parreira, 2006*). Additionally, significant differences in visual analog scale scores indicated greater perceived pain during these movements among patients with migraines compared with those with tension-type headaches (*Martins, Gouveia & Parreira, 2006*). These results align with our study's findings, which also demonstrate higher fear and difficulty scores for exercises involving jolting and rapid head displacements.

Migraine pathogenesis is still not known. However, research suggests several processes are involved in migraines: cortical spreading depression, dysfunction of the trigeminal neurovascular pathways, peripheral and central sensitization, and microglial activation (*Bernstein & Burstein, 2012*; *Noseda & Burstein, 2013*; *Goadsby et al., 2017*; *Su & Yu, 2018*; *He et al., 2023*). Thus, sensitization of the peripheral trigeminovascular neurons that innervate meninges is often present during migraine attacks (*He et al., 2023*). This process might facilitate a heightened pain response to sudden sharp head movements, such as jolting, or to vibrations transmitted during impact activities. Jolting inherently occurs during running and jumping, which could explain the perceived fear associated with these activities in this study. Additionally, the supine position has also been shown to worsen migraine headaches, possibly due to an increase in intracranial pressure (*Doepp et al., 2003*; *Chou et al., 2004*). The spine extension exercise was performed in a Roman chair, reaching the horizontal plane at the beginning of the repetition and extending until almost the vertical plane at the highest point. Hemodynamic changes during forward bending, such as in spine extension, could also explain a possible increment in migraine pain, leading to the

heightened perceived fear and difficulty shown in the results (*Doepp et al., 2003*; *Chou et al., 2004*; *Martins, Gouveia & Parreira, 2006*).

Another important contributor to the fear and avoidance of exercise in the specific case of patients with vestibular migraine could be the potential role of vertigo-inducing exercises. Perhaps, exercises involving spinal movement, particularly movements of the neck and head, could be perceived as exacerbating factors for neck pain and vertigo episode triggers. This perception is significant, as neck pain and the risk of pain intensifying with exercise practice were considered main barriers to exercise rehabilitation adherence in a qualitative study evaluating patients with vestibular migraine (*Kalderon et al., 2024*). However, lower levels of physical activity may be associated with a poorer vestibular migraine prognosis (*Xiao, Bi & Zheng, 2019*). Additionally, the inclusion of an exercise intervention has been shown to improve vestibular migraine symptoms compared to a relaxation intervention (*Sun et al., 2022*), highlighting the relevance of exercise and physical activity in this population too.

The regression analyses revealed that both the perceived fear associated with jumping and the perceived difficulty of spine extension were negatively associated with the IPAQ-SF score. More physically active participants appeared to exhibit lower perceived fear with jumping and lower perceived difficulty with spine extension. However, due to the cross-sectional design of the study, no causal inferences can be drawn. Possible explanations for these findings include that patients with higher physical activity levels may be more familiar with these exercises, have greater tolerance for physical effort due to long-term exercise and subsequent improvements in their migraine symptoms, or have a better baseline migraine status that facilitates exercise tolerance. Previous studies have shown that lower levels of physical activity are associated with a stronger expectation that physical activity could trigger and worsen migraines, a higher frequency of migraine episodes, and consequently, chronic migraine, although no casualty can be confirmed (*Benatto et al., 2017*; *Farris et al., 2019*). Systematic reviews, clinical trials, and cohort studies suggest that exercise might reduce migraine symptoms and disability (*Seok, Cho & Chung, 2006*; *Woldeamanuel & Cowan, 2016*; *La Touche et al., 2020*, *2023*; *Hagan et al., 2021*; *Wu et al., 2022*). The patient's perception of exercise as threatening or difficult could pose a considerable barrier to adherence, thus limiting its effectiveness in reducing migraine impact.

Similarly, both the perceived fear of jumping and the perceived difficulty of spine extension were positively associated with the FAB-Q score. Patients with higher fear-avoidance beliefs appear to perceive greater fear-related jumping and increased difficulty with spine extension exercises. This could suggest that patients with negative past exercise experiences have developed stronger fear and avoidance behaviors toward exercise. This aligns with the associations observed with physical activity levels in the IPAQ-SF, indicating that such beliefs and behaviors might act as a barrier to increasing physical activity, implementing exercise routines, and improving migraine status. Therapeutic education approaches that address these barriers—by challenging negative beliefs about exercise, emphasizing its health and migraine-related benefits, and gradually exposing patients with migraine to it—could be effective in the treatment of this condition.

Biobehavioral interventions, such as those proposed by *La Touche et al. (2024)*, may offer a promising strategy to achieve these objectives by integrating therapeutic education and motivational interviewing (*La Touche et al., 2024*). Within this model, it is important to teach patients to differentiate between physical activity as a migraine trigger and as a worsening factor. Patients should understand that while exercise could worsen migraine pain during migraine phase, it does not necessarily mean it will always trigger an attack. Another key point is to explain that chronic exposure to exercise could diminish the overall impact of migraine, as supported by current evidence.

Moreover, the combination of exercise with cognitive training could interact, decreasing sensory impairments, as suggested by recent findings (*Deodato et al., 2024*). Exercise could reduce the sensitization of the trigeminovascular system and improve cognitive functioning through the release of brain-derived neurotrophic factor (*Moreira et al., 2018*). Therapeutic education could also be crucial in restoring cognitive functioning by increasing exercise adherence and including cognitive tasks in the intervention.

As with the IPAQ-SF score, self-efficacy levels assessed with the CPSES were negatively associated with perceived difficulty in the spine extension exercise, indicating that patients with lower self-efficacy tended to perceive greater difficulty with this exercise. Similar to exercise itself, self-efficacy has been directly linked to lower levels of disability in patients with migraine (*Woldeamanuel et al., 2020*). High self-efficacy is crucial for incorporating exercise-related behaviors and sustaining adherence to them. Biobehavioral interventions provide patients with self-management tools that enhance their control over their condition, thereby boosting self-efficacy (*La Touche et al., 2024*).

Lastly, correlation analyses revealed a moderate negative correlation between the IPAQ-SF score and fear of running. This relationship aligns logically with the association between the IPAQ-SF and fear of jumping, given the similarities between running and jumping in terms of impact, jolting, and head movements. Participants with higher levels of physical activity perceived less fear during running, as was the case with jumping. Although other significant correlations were weak, most indicated that perceived fear during the exercises—especially running, jumping, spine flexion, and spine extension—increased in direct relation to TSK-11, FAB-Q, PCS, and HIT-6 scores. Conversely, CPSES and SEQRE scores were negatively correlated with perceived fear during exercises. A similar trend was observed for perceived difficulty with the exercises. Using these tools to evaluate patients before initiating an exercise program could help tailor the approach to each patient's capacities and perceptions, enhancing their tolerance and adherence to exercise and increasing the likelihood of positive outcomes.

## Clinical implications and future directions

Understanding the levels of physical activity, kinesiophobia, catastrophizing, fear-avoidance beliefs, headache-related disability, and self-efficacy in patients with migraine can be especially beneficial during the initial evaluation and treatment planning, particularly when considering therapeutic exercise interventions within a biobehavioral framework. Knowing these patients' physical condition and exercise perceptions could help healthcare professionals tailor exercise modalities and prescription parameters to each

patient's abilities and perceived sense of safety, promoting a more personalized intervention.

Moreover, considering that various psychological and functional variables, such as self-efficacy and physical activity levels, have been associated with a worse migraine condition, and that increasing the amount of physical activity may directly impact symptoms, studying the relationship between these exercise perceptions and migraine symptoms in prospective and experimental studies could help clarify whether these perceptions influence migraine frequency, intensity, and duration. Perhaps, decreasing the anticipatory anxiety response triggered by exposure to these exercises could reduce migraine-related symptoms.

Furthermore, the findings of this study provide a foundation for designing graded exposure programs for exercise. Given that certain exercises were perceived as more fear-inducing or challenging, these results can help establish a progressive exposure hierarchy. Initiating treatment with exercises perceived as less intense, such as walking, and gradually progressing to more challenging exercises could reduce initial fear and improve adherence. This approach allows for a gradual reduction in psychological barriers, facilitating patient adaptation to exercise levels that could ultimately benefit their condition.

Implementing a biobehavioral intervention that integrates therapeutic education and exercise could also help address these psychological factors, enhancing exercise adherence and improving its overall effectiveness. Therapeutic education could be instrumental in modifying misconceptions and strengthening patients' confidence in their ability to engage in physical activities without fearing migraine exacerbation.

## Limitations

This study presents several limitations that should be considered when interpreting its results and guiding future research.

Firstly, the lack of randomization in the order of exercise presentation is a significant limitation. The assessment of one exercise could have influenced the perception of fear or difficulty in subsequent exercises, particularly those perceived as more intense or challenging. Randomizing the order of presentation would have minimized this carryover effect and enhanced the validity of the findings, allowing for a more accurate analysis of each exercise's perception independently.

Additionally, modifying the FAB-Q questionnaire limited its original scope. However, the internal consistency analysis of the adapted version demonstrated good reliability, supporting its applicability in this population. Nonetheless, the adaptation could have affected comparability with previous studies that used the full version. In addition, although the IPAQ-SF questionnaire is a valid tool to evaluate the level of physical activity, its accuracy in estimating the total METs per week is limited by participants' interpretation regarding each intensity category and their ability to recall the time they spent doing exercise. However, due to the cross-sectional design and online evaluation, using more precise instruments to evaluate the amount of physical activity, such as accelerometers, was not feasible.

Another important limitation is the use of a unidimensional scale to evaluate perceived difficulty and fear. This approach could have restricted measurement sensitivity, with potential "floor" or "ceiling" effects that limit the ability to capture more nuanced variations in patients' perceptions. Employing a multidimensional scale in future studies could provide a more detailed understanding of the factors contributing to perceptions of difficulty and fear across various exercises.

The sample selection also presents a limitation, because it was primarily conducted through online and social media channels. This could have introduced selection bias, potentially excluding patients who might exhibit higher levels of disability and lower exercise tolerance due to a more advanced severity of their condition. Therefore, findings might not be fully generalizable to the broader population of migraine patients in clinical settings, highlighting the need for future studies to include more representative samples across different contexts. In fact, the disproportionate number of females compared to males impeded any stratified analysis or comparison based on the sex category. This could be considered a limitation raised by the non-probabilistic convenience sampling strategy. However, updated evidence shows that women have 3.25 times more risk of developing migraine than men, and even in countries like Italy the female-to-male ratio is 4.9:1 (*Allais et al., 2020*). Considering these data, the present study is almost in alignment with the proportion of women and men affected by migraine.

Lastly, its cross-sectional design limits the ability to establish causal relationships between variables. This design constrains the ability to determine whether physical activity levels, kinesiophobia, and other psychological and clinical factors are causes or consequences of perceived fear and difficulty during exercises. Longitudinal studies are needed to explore the directionality of these associations and provide a foundation for interventions aimed at modifying long-term risk factors. Moreover, non-evaluated factors, such as sleep quality, migraine frequency, intensity, duration, and medication use, could have influenced the model, explaining the high variability in some analyzed outcomes. However, controlling these confounders would need a more complex design. Future studies should prospectively evaluate the effect of these variables on exercise perception.

## CONCLUSIONS

Jumping, running, and spine extension were perceived as the most aversive exercises among patients with migraines, whereas spine extension, jumping, running, squatting, and shoulder press were perceived as the most difficult exercises. The perceived fear score for jumping, identified as the most aversive exercise, was negatively associated with the IPAQ score and positively associated with the FAB-Q score. Similarly, the perceived difficulty score for spine extension, considered the most difficult exercise, was negatively associated with the IPAQ and CPSES scores and positively associated with the FAB-Q score. Weak to moderate correlations were found between the perceived fear and difficulty of most exercises and the questionnaire scores, highlighting the relevance of evaluating physical activity, kinesiophobia, catastrophizing, fear-avoidance beliefs, headache disability, and

pain and exercise self-efficacy levels in patients with migraine before initiating a therapeutic exercise program.

## ACKNOWLEDGEMENTS

We would like to express our gratitude to Irene Sánchez Ruiz for performing all the exercises featured in the videos used in this study. Our sincere thanks also go to Rubén Fernández Matías for his invaluable expertise and significant assistance with the statistical analysis of the data.

### Funding

This article was funded by internal financing from Centro Superior de Estudios Universitarios La Salle, with the code GI2012002C, in 2023. The funders had no role in study design, data collection and analysis, decision to publish, or preparation of the manuscript.

### Grant Disclosures

The following grant information was disclosed by the authors:
Centro Superior de Estudios Universitarios La Salle: GI2012002C.

### Competing Interests

The authors declare that they have no competing interests.

### Author Contributions

- Álvaro Reina-Varona conceived and designed the experiments, performed the experiments, analyzed the data, prepared figures and/or tables, authored or reviewed drafts of the article, and approved the final draft.
- Beatriz Madroñero-Miguel conceived and designed the experiments, prepared figures and/or tables, authored or reviewed drafts of the article, and approved the final draft.
- Alba Paris-Alemany conceived and designed the experiments, authored or reviewed drafts of the article, and approved the final draft.
- Roy La Touche conceived and designed the experiments, analyzed the data, authored or reviewed drafts of the article, and approved the final draft.

### Human Ethics

The following information was supplied relating to ethical approvals (*i.e.*, approving body and any reference numbers):

The study obtained the approval of the Ethics Committee of the University La Salle (CSEULS-PI-005/2022).

### Data Availability

The raw data measurements are available in the Supplemental File.

## Supplemental Information

Supplemental information for this article can be found online at http://dx.doi.org/10.7717/peerj.19342#supplemental-information.

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
