# Peer review of "Perceived fear and exercise difficulty in patients with migraine and their association with psychosocial factors: a cross-sectional study"

_PeerJ, doi:10.7717/peerj.19342_

## Round 0.1 · original submission · Major Revisions

Dear Authors

The referees have major concerns with the paper. Kindly respond, revise and resubmit.

·

Basic reporting

Dear authors,
First of all, thank you for the invitation to review your study “Perceived fear and exercise difficulty in patients with migraine and their association with psychosocial factors: A cross-sectional study”. The following suggestion could improve the quality of your study, please see the specific suggestions below.


Please replace the term patient with subject
Revise the references section with more recent references
The quality of tables and figures should be improved

Experimental design

• The timeline of the study should be added
• Details concerning the date of the approval of institutional review board should be added.
• The manuscript does not clearly justify the chosen sample size or discuss how it ensures sufficient power to detect expected results.
• Were the evaluation and exercise performed in the pain free days?
• In which phase of menstrual cycle were the female subjects evaluated?

Validity of the findings

• I would add a more comprehensive discussion of the study's limitations, including the impossibility make a gender stratification.
• The discussion would benefit from more detailed suggestions for future research, including potential effect of patients education

Additional comments

Introduction
• The pathophysiology and the burden of migraine be introduced in order to support your topic.
• Regard to the paragraph of physical activities in migraine, there are several studies more recent than the ones you mentioned, moreover these studies are also trial and systematic review. Please take in to consideration these articles for introduce the concept of physical activities in migraine:

Deodato M, Granato A, Buoite Stella A, Martini M, Marchetti E, Lise I, Galmonte A, Murena L, Manganotti P. Efficacy of a dual task protocol on neurophysiological and clinical outcomes in migraine: a randomized control trial. Neurol Sci. 2024 Aug;45(8):4015-4026. doi: 10.1007/s10072-024-07611-8. Epub 2024 May 29. PMID: 38806882; PMCID: PMC11255006.

Lemmens J, de Pauw J, van Soom T, et al (2019) The effect of aerobic exercise on the number of migraine days, duration and pain intensity in migraine: A systematic literature review and meta-analysis. J Headache Pain 20. 10.1186/s10194-019-0961-8

• In order to introduce better your research gap, it would be useful to better described the differences between migraine trigger and migraine contributing factors. In fact, physical activities may act as a migraine trigger when the patients are in the prodrome phase, just before the migraine attack, or physical activities may worse the intensity of the migraine attack when the patients are in the migraine attack phase. On the other hand, physical inactivity may cat as contributing factor of migraine. In fact, Active exercise act with the release of the brain-derived neurotrophic factor that promotes cognitive performance and neuroplasticity, but also active exercise may act with the reduction of central sensitization that restores cognitive functioning. In this line patient’s education play a pivotal role. Please explain this concept in order to introduce your research gag and the aim of your study.

Reviewer 2 ·

Basic reporting

.

Experimental design

.

Validity of the findings

.

Additional comments

Overall Assessment:
This manuscript presents a well-designed cross-sectional study that explores the perceived fear and difficulty of various exercises in patients with migraine. The study addresses an important gap in the literature, providing valuable insights into the psychological factors that influence exercise adherence in this population.

Strengths:

1. Clear research question: The study objectives are well-defined, and the research question is relevant to the field.
2. Methodological rigor: The study employs a robust methodology, including a suitable sample size, valid outcome measures, and appropriate statistical analyses.
3. Clinical significance: The findings have important implications for the development of personalized exercise interventions for patients with migraine.
4. Well-organized and clearly written: The manuscript is well-structured, and the writing is clear and concise.

Weaknesses and Suggestions:

1. Limited generalizability: The study sample consists mainly of women, which may limit the generalizability of the findings to other populations.
2. Cross-sectional design: While the study provides valuable insights, the cross-sectional design limits the ability to establish causality between exercise perceptions and migraine symptoms.
3. Measurement tools: Although the study employs validated outcome measures, some tools (e.g., IPAQ-SF) may have limitations in terms of accuracy or responsiveness.
4. Lack of consideration of confounding variables: The study does not appear to control for potential confounding variables that could influence the relationship between exercise perceptions and migraine symptoms, such as sleep quality, physical activity level, migraine frequency and severity, comorbidities, and medication use.
5. Vertigo-inducing exercises: The study does not explicitly discuss the potential role of vertigo-inducing exercises in contributing to the fear and avoidance of exercises in patients with migraine.

Conclusion:
This manuscript presents a well-designed study that contributes to our understanding of exercise perceptions in patients with migraine. However, the study has some limitations, including the lack of consideration of confounding variables and the limited generalizability of the findings. With some revisions to address these weaknesses, this manuscript has the potential to make a significant contribution to the field.

Recommendation:
I recommend that the authors revise the manuscript to address the weaknesses mentioned above. Specifically, they should:

1. Discuss the limitations of the study sample and potential strategies to increase diversity in future studies.
2. Consider adding a discussion of potential causal relationships between exercise perceptions and migraine symptoms.
3. Provide more detail on the measurement tools used and their limitations.
4. Assess and control for potential confounding variables, such as sleep quality, physical activity level, migraine frequency and severity, comorbidities, and medication use.
5. Discuss the potential role of vertigo-inducing exercises in contributing to the fear and avoidance of exercises in patients with migraine.
6. Expand the discussion of future research directions, including potential interventions to address exercise perceptions and adherence in patients with migraine.

With these revisions, I believe the manuscript will be ready.

---

## Round 0.2 · accepted · Accept

Dear Authors

Thank you for addressing all the concerns of the referees who have now recommended that the paper be accepted

Congratulations

·

Basic reporting

The article looks improved. I thank authors for the work done.

Experimental design

ok

Validity of the findings

ok

Reviewer 2 ·

Basic reporting

clear

Experimental design

robust design

Validity of the findings

valid findings

Additional comments

none